# In the SARS-CoV-2 Pandora Pandemic: Can the Stance of Premorbid Intestinal Innate Immune System as Measured by Fecal Adnab-9 Binding of p87:Blood Ferritin, Yielding the FERAD Ratio, Predict COVID-19 Susceptibility and Survival in a Prospective Population Database?

**DOI:** 10.3390/ijms24087536

**Published:** 2023-04-19

**Authors:** Martin Tobi, Martin H. Bluth, Noreen F. Rossi, Ereny Demian, Harvinder Talwar, Yosef Y. Tobi, Paula Sochacki, Edi Levi, Michael Lawson, Benita McVicker

**Affiliations:** 1Research and Development Service, Detroit VAMC, 4747 John R Street, Detroit, MI 48602, USA; 2Blood Transfusion and Donor Services, Department of Pathology, Maimonides Medical Center, 4802 10th Avenue, Brooklyn, NY 11219, USA; 3School of Medicine, Wayne State University, 540 E Canfield St, Detroit, MI 48201, USA; 4Division of Nephrology, Department of Physiology, School of Medicine, Wayne State University, 540 E. Canfield Ave., Detroit, MI 48201, USA; 5Department of Internal Medicine, Pennsylvania State University College of Medicine, 700 HMC Cres Rd., Hershey, PA 17033, USA; 6Department of Thoracic Oncology, Memorial Sloan Kettering Cancer Center, 1275 York Ave, New York, NY 10065, USA; 7Department of Pathology, Detroit VAMC, 4747 John R Street, Detroit, MI 48602, USA; 8Division of Gastroenterology and Hepatology, University of California at Sacramento, Sacramento, CA 95819, USA; 9Research Service, VA Nebraska-Western Iowa Health Care System, Omaha, NE 68105, USA; 10Department of Internal Medicine, University of Nebraska Medical Center, Omaha, NE 68198, USA

**Keywords:** COVID-19, innate immune system, adaptive immune system, p87, Adnab-9, organ specific neoantigen, prognostic biomarker, Abl, Src, ACE inhibitors, lysozyme, diabetes mellitus, race, sex

## Abstract

SARS-CoV-2 severity predictions are feasible, though individual susceptibility is not. The latter prediction allows for planning vaccination strategies and the quarantine of vulnerable targets. Ironically, the innate immune response (InImS) is both an antiviral defense and the potential cause of adverse immune outcomes. The competition for iron has been recognized between both the immune system and invading pathogens and expressed in a ratio of ferritin divided by p87 (as defined by the Adnab-9 ELISA stool-binding optical density, minus the background), known as the FERAD ratio. Associations with the FERAD ratio may allow predictive modeling for the susceptibility and severity of disease. We evaluated other potential COVID-19 biomarkers prospectively. Patients with PCR+ COVID-19 tests (Group 1; n = 28) were compared to three other groups. In Group 2 (n = 36), and 13 patients displayed COVID-19-like symptoms but had negative PCR or negative antibody tests. Group 3 (n = 90) had no symptoms and were negative when routinely PCR-tested before medical procedures. Group 4 (n = 2129) comprised a pool of patients who had stool tests and symptoms, but their COVID-19 diagnoses were unknown; therefore, they were chosen to represent the general population. Twenty percent of the Group 4 patients (n = 432) had sufficient data to calculate their FERAD ratios, which were inversely correlated with the risk of COVID-19 in the future. In a case report of a neonate, we studied three biomarkers implicated in COVID-19, including p87, Src (cellular-p60-sarcoma antigen), and Abl (ABL-proto-oncogene 2). The InImS of the first two were positively correlated. An inverse correlation was found between ferritin and lysozyme in serum (*p* < 0.05), suggesting that iron could have impaired an important innate immune system anti-viral effector and could partially explain future COVID-19 susceptibility.

## 1. Introduction

The world has been reeling from the onslaught of the COVID-19 pandemic, with the vast disparity between those susceptible to infection and those who have been infected with poor outcomes. Despite the availability of vaccines, COVID-19 infections continue to be an epidemiological challenge. Knowledge of susceptibility would be informative, so that the vaccination effort could focus on those at the greatest risk of infection. To clarify this primary issue, we deployed a prospective study focusing on the pre-morbid stance of the innate immune system to the novel SARS-CoV-2 (severe acute respiratory syndrome-coronavirus 2) agent. Most studies have been retrospective and descriptive, as described below, making predictive biomarkers difficult to detect while leaving the science dependent on retrospective epidemiological studies for insight. There have been few truly pre-morbid, prospective COVID-19 studies published, and most have been from the UK medical system [1,2,3,4,5,6,7] and have addressed mental health, neurological deficits, HIV, and socioeconomic issues. Most have shown convincing positive correlations. There could also be others not classified as such [8], but none have appeared to focus directly on the pre-morbid innate immune system. In terms of the general pandemic literature, we briefly summarized the approximate percentage of the original research contributions and commentary for the past 3 coronavirus epidemics, including the ~91 COVID-19 retractions, some of which had been in top medical journals, and this resulted in a loss of faith and trust [9] in present and ancient times [10]. At the time of this report, there have been approximately 140,252 COVID-19 articles written and 91 retractions, as compared to 0 retractions for the SARS and MERS pandemics. Most of these articles were original works.

While studying a Paneth-cell marker for colorectal neoplasia (CRC) prospectively [11], we chanced upon patients who had contracted COVID-19, and thus, we had been able to associate the pre-morbid stance of a marker using a simple ratio of the blood concentration of ferritin as the numerator and the stool p87 ELISA results as the denominator, which appeared to correlate with the previously acquired COVID-19 disease. The current thought is that the initial host response to the SARS-CoV-2 is orchestrated by the InImS with at least partial success in containing the virus during the viral-replicative phase, which then presumably causes only mild disease. Following this, the adaptive immune system is activated, accompanied by the production of antibodies via a vigorous inflammatory response, typically described as a “cytokine storm”, which, potentially, can be lethal. Recent evidence has suggested that host antibody-mediated inflammatory cell death of monocytes by a specific FcγR (Fc gamma receptor recognition of immunoglobulin IgG) could lead to overwhelming inflammation [12]. Those at increased risk for this scenario have typically been older patients and those with underlying systemic disease [13], but previously healthy individuals have also displayed severe responses to SARS-CoV-2. Treatment has been directed at the virus, and the efficacy of such medications, other than corticosteroids [14], which is still debatable, has been modest, at best, as compared to vaccination effectiveness, in preventing mortality. However, even for these, supply logistics and population acceptance can be challenging. We reviewed a variety of potential biomarkers of the InImS [15] and correlated these with susceptibility and severity factors.

## 2. Results

### 2.1. Patient Accrual

The participants totaled 2292 with the breakdown depicted in Table 1 and Figure 1. For the actual FERAD ratios available from a colorectal cancer screening study (NCT01815463), wherein the patient-collected stool samples on commercial cardboard FOBT cards were delivered via postal mail. A total of 270 patients successfully participated in Phase 2 (12%), which was within the target range of 10–15% of all patients enrolled. Overall, we achieved a successful enrollment of 82% of our target number. Of the participants, 12% were female, which exceeded our goal of 10%, and this exceeded the current proportion of 9% in 2014 [16], as compared to 50% of the general population.

### 2.2. Specimen Collection and Diagnostic Procedures in Test and Control Populations

We obtained stool p87 and ferritin in 10 of the 28 patients in Group 1 (positive symptoms and PCR test for COVID-19) at the Detroit VAMC, or elsewhere; in 13 of the 46 patients in group 2, who were the controls with COVID-19 compatible symptoms at screening but who had tested negative for COVID-19 or had the absence of specific antibodies; in 23 of the 90 patients in Group 3, who had negative test results and no symptoms; and in 461 of the 2129 patients in Group 4, whose testing and symptoms were unknown. The diagnostic test was the Cepheid PCR (see Table 1) in most instances, and many of the patients had repeated testing to gain admittance to clinics, for scheduled outpatient procedures, and before clinically relevant inpatient patient transfers. The recovery from disease was defined as a later COVID-19-negative test or the resolution of all symptoms. Of these patients, 22% had both study-directed p87 and opportunistic ferritin levels available (due to only 44% of the patients submitting stool cards and only 50% having had at least 1 ferritin level drawn; −50% of 44% was 22%) to enabled the derivation of their FERAD ratios. The demographics and the duration of admission/disease are also shown in Table 1 and Table 2.

Having compared the groups of COVID-19 patients, we now compared their clinical features and disease outcomes between groups 1 and 2.

Table 2 provides details of the clinical parameters and the outcomes in COVID-19 patients (group 1) and the controls (group 2), as well as a comparison with a COVID-19 cohort from the first wave of the pandemic.

Detroit veterans had experienced two waves of COVID-19 illness. More in-depth information with a comparison to the earlier first wave in early March 2020 had been published [17] as a descriptive paper, and the results matched the data in Table 2. It was important to compare patients from the 2020 wave to our combined COVID-19 patient cohort (group 1) to find significant differences that could have resulted from the benefits of medical and nursing care that had improved with increased experience, and this was, indeed, the case. Therefore, we validated our sample with the initial published sample [17].

Despite the temporal differences between these 2 groups of COVID-19 patients, when the comparison data were correlated between the 2 groups, there was a convincing positive correlation with a correlation coefficient of r = 0.995 that was statistically significant at the *p* < 0.00001 level in Figure 2.

### 2.3. Ferritin, Lysozyme Levels, and p87 ELISA Estimations

Comparatively few of the 2 small patient clinical populations sampled had turned in stool samples: 44% from COVID-19 group versus 46% from the controls. Despite this limitation, it was found that the historic values of ferritin were significantly higher in those destined to have the disease, as compared to those who did not (420 ± 84 versus 122 ± 173 ng/dL; *p* < 0.018). The ferritin in patients with diabetes was similar. Specifically, the initial historic blood ferritin levels were correlated with last drawn levels in 56 patients, as shown in Figure 3a, with r = 0.41 (*p* < 0.002).

As compared to the first and last recorded ferritin levels, we were able to show in 56 COVID-19 patients, group 1, and controls (group 2 and 3) that the relationship between these 2 parameters had been conserved over a prolonged time period. The mean ± standard deviation of the time between the initial and last ferritin level measurements were 9.8 ± 5.9 years. Of the sampling, 34% in Groups 1, 2, and 3 had at least 2 ferritin results available.

Figure 3a illustrates the tight positive correlation between the first and last measured ferritin, suggesting an exact homeostasis maintained by the body to maintain constant body iron stores. Having confirmed a relationship between the historic and most current ferritin levels in Figure 3a, we considered other laboratory tests that might correlate with the ferritin blood levels. We then plotted the historic ferritin levels against those of the stool p87, though these had not necessarily been collected at the same time. The results are depicted in Figure 3b. The interval from the time of the drawing of the ferritin samples was 12.8 ± 10.6 years, and for Adnab-9, 17.8 ± 2.3 years. This difference was not significant (*p* > 0.05).

The scattergram in Figure 3b shows a positive correlation before the two parameters needed for determination of the FERAD ratio, suggesting a close relationship between the iron stores (blood ferritin) and the innate immune system product of intestinal Paneth cells (p87).

Paneth cells are an integral part of the innate immune system, and p87 was a known Paneth-cell product [11,18]. This represented a relationship between this aspect of the immune system and the intracellular iron stores [19]. We sought to explore this relationship by obtaining a ratio of the ferritin expressed in ng/dL and the fluorescence optical density of Adnab-9 in OD-minus-background/5 µg protein per well (FERAD), in both the patients and the controls.

### 2.4. FERAD Ratios and Lysozyme

Having shown a direct correlation between p87 and ferritin, we were able to relate the FERAD ratio (Ferritin in ng/dL divided by Adnab-9 stool-binding in OD/5 ug protein-background) to the COVID-19 and control cohorts, as shown in the analysis in Figure 4.

The mean FERAD ratio levels in Group 1 COVID-19+ patients were 6815 ± 7823 versus 69,575 ± 132,845 in Group 2 controls (*p* < 0.05); 14,608 ± 31,263 in the Group 3 patients with negative tests and no COVID-19 signs (NSS); and in Group 4, where there was no available information about tests or signs, FERAD differences were significant at 50,704 ± 114,298 (*p* = 0.000002). Data were available for 39% of the COVID-19 patients, 41% in (Group 2) controls, and 11% negative and 20% in the control (Group 3) and negative test (Group 4) groups, respectively. The results are depicted in Figure 4.

Despite significantly low FERAD levels in the COVID-19 group with a relatively low standard deviation, the control groups had significantly higher FERAD ratios with a much larger range of values, suggesting that the datasets of the controls had a greater variation in the ferritin blood values and fecal p87 estimations and were not as homogeneous as COVID-19 patients, who could have been at risk because of their low FERAD ratios. We also performed the test a second time using 10 COVID+ FERAD patient results and, as expected, received a slightly higher *p* = 0.05 value.

Paneth cells also secrete lysozyme (neuraminidase), a low molecular weight anti-microbial molecule that has also been shown to have antiviral properties, including against SARS-CoV-2 [20]. Figure 5 shows, as expected, the iron degradation of lysozyme [21], resulting in an inverse correlation between the lysozyme and ferritin in 18 patients, where only 1 patient contracted COVID-19 disease in the future.

We were interested in comparing the FERAD scores in our entire surrogate patient database against the various groups at lower or higher risk of contracting the disease and having potentially worse outcomes.

Of interest, the FERAD ratios in the non-COVID-19 patients in the database showed that elderly patients over 60 years of age tended to have greater mean FERAD ratios than younger patients (55,783 ± 128,346 versus 40,017 ± 82,311, respectively, a trend at *p* = 0.1). Females had significantly lower FERAD levels than males (23,207 ± 44,131 versus 53,940 ± 119,412, respectively; *p* < 0.008). This could explain the observation that males had more severe disease and mortality [22], similar to the aforementioned ferritin levels that were a predictive biomarker for severe infection. In order to further clarify the FERAD contribution, we examined the demographic data of underlying clinical diseases associated with worse COVID-19 outcomes and their contributions to overall mortality, as compared to the more specific p87 testing. Table 3 depicts the FERAD scores from the CRN database.

Table 3 summarizes the FERAD scores in relationship to the demographic features and the presence of diabetes. In this particular analysis, only sex differences were significantly different in the FERAD ratios; however, the footnotes showed a better survival than males, possibly based on the age differences. Having discovered the association of the FERAD ratio to disease susceptibility, we focused on the possibility of predicting overall mortality. We were careful to ensure that confounders, such as age, did not unduly affect our predictors and used quartiles based on the FERAD ratios to exclude the effect of confounders. The results are shown in Figure 6.

Differences in survival were observed when comparing the extremes within each quartile that were significant in all survival differences, except when considering the shed p87 in the effluent by ELISA. No survival rates for African Americans had been recorded in quartiles 3 and 4, as there were no p87 bands present in these quartiles. This could have suggested an adverse prognosis in the absence of specific bands. The applicable odds ratios and the confidence intervals for while-integers of the *p*-values, from left to right, in the figure are 2.73 (1.30–5.71); *p* < 0.012; 0.41 (0.17–0.98); *p* < 0.045; 2.04 (1.15–3.60); *p* < 0.016, respectively. There were no significant differences in survival based on effluent ELISA results, but there were significant differences in ELISA results between the quartiles (see Table 4).

By applying the aforementioned strategy to the surrogate group in which the COVID status was unknown in the living participants but did include many deceased patients, as opposed to those who had COVID-19 testing in the recent waves of the pandemic, We were able to determine the mortality by the FERAD quartiles, as previously mentioned.

The FERAD quartiles enabled the differentiation of the demographic and effluent changes. The demographic changes were not significant, but significant changes in the elevated effluent p87 could be observed in quartiles 2 and 4, suggesting that the higher FERAD values increased the effluent p87 at both the lower and upper interchanges. There was likely a direct relationship between the FERAD values and the InImS at these junctures.

There were no correlations between p87 in the stool or the effluent in 245 paired samples (*p* = 0.91) for the quartiles, but there was an inverse relationship between these parameters in Group 2 patients, who were symptomatic but had COVID-19 negative tests (See Figure 6 above). No correlations were observed between the effluent p87 and the OSN ELISA results in 64 paired samples (*p* = 0.298) or the stool p87 and the OSN in 61 paired samples (*p* = 0.22). An inverse relation between the stool and the effluent p87 was observed (see Figure 7 below) in Group 2 patients (r = −0.87; *p* < 0.012).

We then explored the relationship of various p87 manifestations, such as the specific bands in western blotting, the bound tissue p87 in the various regions of the colon, and the relationship between the shed stool p87antigen and that in the colonic effluent, which were the washings obtained during the index colonoscopy, particularly given the differences observed between African American and Caucasian populations in the different control groups.

The differences could have been explained by the perturbations of the Paneth cells with the lower shed stool p87 before colonoscopy, increasing that of the effluent caused by the stress of preparing the bowel and the colonoscopy procedure itself.

### 2.5. Immunohistochemistry

In order to shed light on the differences in COVID-19 mortality rates in AA and Caucasian populations, we performed a semi-quantitative immunohistochemistry (either positive or negative) on colonoscopically obtained biopsies from 6 colonic regions using the Adnab-9 antibody that defined p87. Figure 8 shows a representative example of positive Adnab-9 binding.

We endeavored to reveal the quantitative differences between these regions by ethnicity in 112 AA and 100 Caucasians. We found that although the right colon regions generally had an increased p87 expression, as compared to the left, the expression was statically significantly decreased in the ceca of AA (OR 0.43 [CI 0.24–0.78]; *p* < 0.008), as depicted in Figure 9. This likely reflected the distribution of the colonic Paneth cells in the two ethnic groups, which had been compared, and could have been linked to a JAK3 mutation more frequently found in African Americans [23].

Figure 9 shows the significantly lower levels in the mean p87 results in African Americans and Caucasians, with a significant difference in the cecum (*p* < 0.008) of the former. The cecum was most likely from other right-sided colonic segments (ascending and transverse colon) to harbor a population of Paneth cells in the normal colon.

We also found a correlation between p87 and other indices of iron content, such as iron saturation (r = 0.5; *p* < 0.025, as depicted) confirming the importance of iron to the innate immune system. In contrast to the latter finding, there was no correlation between p87 and OSN, suggesting that they could represent different components of the immune system. In the surrogate group 3, there was an inverse correlation between the p87 ELISA and the ferritin (r = −0.1; *p* < 0.028), which was consistent with an inverse relation of the ferritin and ELISA p87 correlation (Figure 10) in this larger cohort. Since we had not correlated the Western blotting data on the stool, we could not determine the significance of this finding.

### 2.6. The Western Blotting, Colonic Effluent, Severity of Disease Prediction by FERAD, and Absolute Neutrophil/Absolute Lymphocyte Ratio

Focusing on the qualitative aspects of p87 in the surrogate database and the relation to the medications, whose efficacy was still unclear within the context of COVID-19, we summarized our approach. The first group for analysis was predicted regarding specificity, as we had postulated that veterans with distinct antigenic bands on Western blotting would reflect an intact immune system, as p87 was a Paneth-cell secretory product. A total of 69 patients had a negative ELISA, yet the specific bands were evident (WB+), and these were compared to the outcomes of 228 patient controls with positive ELISA results but with no detectable bands (as shown in Figure 11). Their mortality rates were lower in the WB+ than in the ELISA positive group (30 versus 51%, respectively; (OR 0.42 [CI 0.23–0.77]; *p* < 0.007). The Paneth cells also act as natural anti-microbials, and we had previously investigated one such entity, alpha-defensin 5 (aD5), that was often co-localized with p87 in various tissues and metaplastic epithelium [18]. This could play an extremely important role in COVID-19 viral infectivity, on which we elaborate in the discussion section. The overall mortality rates were higher, where specific p87 fecal bands were not detected.

Having established a putative survival advantage based on specific p87 stool bands, we wanted to find potential positive associations with the various medications in the surrogate group in order to determine if a plurality had been taking aspirin (ASA), as compared to the group with ELISA positive stool yet with specific bands absent. ASA was prescribed to 41.1 percent of WB+ individuals at the time of their stool collection versus 25.9 percent of the controls (OR 2.0 [CI 1.04–3.83]; *p* < 0.041), suggesting a potential role for ASA in survival. We then turned to the surrogate populations and were able to define a group taking ASA. In a separate analysis of aspirin versus non-aspirin prescribed controls, significantly fewer patients had positive p87 ELISA results (15.9 versus 26.8% (OR 0.52 [CI 0.31–0.87]; *p* < 0.014) and had increased WB+ versus controls (19 versus 11.4%: OR 1.81 [CI 1.02–3.21]; *p* < 0.046). Significantly fewer patients with WB+ had low stool protein (<9 µg/mL) versus controls (6% versus 33.9%: OR 31.3 [CI 10.9–89.4]; *p* < 0.0001). WB+ patients were on average ~4 years older than the controls (64.2 ± 12.1 versus 60.7 ± 12 years, respectively; *p* < 0.04). The semi-quantified p87 IHC labeling was significantly reduced in the ascending colon at 0.125 ± 0.289 versus 0.443 ± 0.733 (*p* < 0.02) with strong trends in the cecum (*p* = 0.08) and rectum (*p* = 0.085). There were no significant differences in the proportion of patients prescribed NSAIDs or with habits such as smoking and drinking. We showed that specific bands in the stool were stable over time and detectable (Figure 12). The observed bands were reproducible and stable, even when left at room temperatures for up to a week.

Using a slightly different approach, we ascertained the number of COVID-19 cases in all patients who had positive p87 in stools, in effluent samples, or in both. There were more COVID-19 cases in the positive groups.

Although the FERAD ratio did not require the effluent p87 input, the historic effluent p87 results were generally higher than that of the stool. In p87 positive stool, there were 5 COVID-19 cases, out of 250 (2%), versus only 4 cases testing negative, out of 754 (0.53%) (OR 3.83 (1.02–14.36); *p* < 0.048). The differences between the patients with the positive samples were even greater, with 4 out of 42 positive, 9.5%, versus 1 out of 175 negative samples (OR 16.38 (CI 1.78–150.40); *p* < 0.008). The combined p87 stool and effluent yielded significant *p*-values of 0.03 versus 0.005% cases, respectively, being COVID-19 positive (OR 5.86 (CI 1.95–17.63); *p* = 0.0015).

An important question to be answered was the ability of the FERAD biomarker to predict increased severity of the disease. Having discovered the relationship between the FERAD ratio and disease susceptibility, we focused on the possibility of predicting the overall mortality. We were careful to ensure that confounders, such as age, did not unduly affect our predictors and used quartiles based on the FERAD ratio to exclude the effects of confounders. We were able to show this (Figure 13) in our limited group of patients and, later, compare our biomarker with another previously published biomarker (see below). The mean FERAD was significantly reduced with mild disease (2039 ± 2382) versus moderate and severe cases (7114 ± 2017), with 4 patients in each group.

Figure 13 is a bar diagram showing a significant difference between the historic FERAD ratios in COVID-19 patients with mild versus severe disease. Despite n = 4 in each group, the FERAD differences according to the severity of disease were significant, suggesting a biomarker role for the FERAD ratio. Therefore, after identifying susceptibility by a low FERAD ratio, we were able to further select those with higher FERAD ratios in order to identify those at increased risk for worse severity of disease.

After reviewing the literature, we found that the neutrophil/lymphocyte ratios (NLR) had also been correlated with disease severity and even mortality [24]. There had been a positive correlation of this ratio between historic levels and the levels at the time of COVID-19 (Figure 14). Not surprisingly, there was also a significant inverse correlation in the control patients (Figure 15) under the same collection conditions.

Figure 15 is a scattergram showing the same ratio of the absolute neutrophil/lymphocyte ratio measured at the time of a negative COVID-19 test, with the ratio at the time of the stool collection. It was statistically significant, but the correlation was inverse and opposite to that of patients with active COVID-19.

These data strengthened the concept that biomarkers taken in the distant past could predict future events. In terms of the innate immune system, NLR had the neutrophil, an innate immune system effector, as the enumerator and the adaptive immune system lymphocyte as the denominator, which was the inverse of the FERAD ratio. Since medications could affect immune function, we considered the medications and their use in COVID-19.

We examined the disease status in the entire group and found that diabetes prevalence was equivalent in both groups, at just over one-third of patients (see below). Since p87 was a neoplasia marker and a potential confounding variable, we wanted to exclude the issue of adenoma detection. There were no differences in the adenoma detection rates at the index or follow-up colonoscopy (~67%) between the ASA-prescribed medication group and those not prescribed.

### 2.7. Neonatal Src and Abl and Relationships to COVID-19

In our case report from an earlier study, we were fortunate to have sample results from birth to infancy from a participant in an earlier study, as well as an available ferritin level from the post-COVID-19 period, when the participant was an adult. The neonatal and early infantile period meconium and stools were extracted and standardized by sample proteins for use in a standard ELISA. Figure 16 shows the binding of the three antibodies with spikes at physiological events (residual umbilical cord shedding) and a minor medical procedure.

The line diagram traced the shedding of the three antigens over time from the neonatal period to early infancy. Figure 16 shows the optical densities (OD) of the ELISA, with p87 being the dominant antigen. The dates are given as month/day/year.

Interestingly, there was a close and significant correlation between p87 and Src but not Abl2 (Figure 17). With additional data, a FERAD score of 248.59 was calculated, which, according to our aforementioned hypothesis, would have indicated that the infant had a high susceptibility to COVID-19.

Accordingly, this patient, as an adult, did contract a relatively severe and prolonged infectious course of COVID-19, from which the patient largely recovered without the need for hospitalization or oxygen support.

### 2.8. Medications and COVID-19

Over the course of conducting a study on the p87 stool tumor marker, which is known to be expressed in gut Paneth cells [25] and a component of the innate immune system, we included patients taking ACE-I or angiotensin-II receptor blockers (ARB) at the time of stool collection. We compared p87 and its broken-down products in the ACE-I (n = 196) and ARB (n = 9) groups (total subset, n = 205), as well as in the controls who had not been on either drug (n = 486). The demographics, as expected, showed older ages (age 65.6 ± 10.16 versus 61.03 ± 11.91 years, *p* < 0.0003) and predominantly male sex (91.7 versus 87.7%; *p* < 0.021), whereas African American patients were similarly distributed (51.2 versus 52.4%; *p* > 0.05) in the ACE-I/ARB group vs. controls, respectively. Diabetes was more common in patients taking ACE-I/ARB than in controls (51.2 versus 31.8%; OR 2.25 (1.61–3.14), respectively; *p* < 0.0001). The stool ELISA for p87 was positive in 24.5 in ACE-I/ARB versus 23.1% in controls (not statistically significant, NSS). Similarly, the more specific Western blotting for all moieties of p87 was 8.9% for ACE-I/ARB versus 13.7% control patients (NSS). In contrast, when considering the more ubiquitous p30 moiety, the proportions were statistically significant (OR 0.43 (0.21–0.89); *p* < 0.021) with positive results in only 5.4% of ACE-I/ARB patients (9 positive, 159 negative) versus 11.7% of controls (48 positive, 362 negative). Since p87 could also be used as a tumor marker and could have been a confounding variable, we also compared the cumulative adenoma numbers in both groups and found that they did not differ (1.94 ± 2.97 in ACE-I/ARB versus controls 2.15 ± 3.64; *p* > 0.05). It was suggested that calcium channel blockers (CCB) could be substituted for ACE-I/ARB for some conditions in individuals at high risk of COVID-19 since they were not known to increase ACE2 [26]. As compared to the p30 proportion of total p87 fecal bands in the Western blotting (see Figure 12 above), the CCB patients had 12.1% versus 13.1% in controls (NSS), suggesting an absence of discernable immunosuppression. Notably, p87 immunostaining was found in the lung tissue as well as the bowel (Figure 18).

### 2.9. COVID-19 and Diabetes Mellitus

A report from India [27] had hypothesized that patients with uncontrolled diabetes could be at risk of re-infection with COVID-19 due to their impaired adaptive immune system aberrations that had been perhaps similar to the aforementioned hypothesis. It appeared that most reports from other countries, such as the United States, China, and Italy, did not support a role for DM in susceptibility [28,29], but the relative proportions of the 4 isoforms of ACE2 at the basis of susceptibility and immune response were not known in these countries. To investigate the DM-related susceptibility and potential mechanisms, we used our characterized high-risk CRC population to study the question. A sizeable proportion of our patients had DM, as determined by the problem list of the VA computerized patient medical record. DM was also a risk marker that had a relationship to colorectal neoplasia, inflammation, and the response to infection. The median follow-up time of the DM population was 10.3 years. A total of 1542 patients were studied, 16 out of 584 with DM, and 9 out of 958 controls had contracted COVID-19 (2.7 vs. 0.9% with OR 2.97 (CI 1.30–6.77); *p* < 0.013). A total of 39.6% DM patients had a FERAD > 15,646 versus 30.8% in controls (1.48 (CI 1.007–2.17); *p* = 0.051). The rectal p87 was elevated in DM 0.360 ± 0.681 versus 0.165 ± 0.401 in controls, *p* < 0.017. The blood ferritin was also elevated in DM (209 ± 229 versus 166 ± 252, *p* < 0.025).

The DM patients were predominantly African American, 57.5 vs. 50.7% (OR 1.32 (CI 1.07–1.62); *p* < 0.011); had more cancer, 40.5 versus 31.3% (OR 1.49 (CI 1.2–1.86); *p* < 0.0005); and had a cumulative adenoma prevalence (2.57 ± 4.5 versus 1.91 ± 3.22; *p* < 0.007). The population was predominantly male (OR 1.47 (CI 1.05–2.07); *p* < 0.031), and HbA1c levels over 6.4 (not solely used for diagnosis of DM) were observed, as expected predominantly in DM patients (74.9% versus 14% in controls (OR 18.4 (CI 12.27–27.43); *p* < 0.0001). Surprisingly, the overall mortality (unrelated to COVID-19) was lower in the DM group (41.6 versus 48.8%, *p* < 0.007), despite DM patients being overweight (BMI > 28, 63.9 versus 41.6%), but this could be attributed to heavier alcohol use in the controls, at only 19.1% in DM versus 31.5% in the controls (OR 0.51 (CI 0.36–0.73); *p* < 0.002). We found an almost 3-fold increased risk of susceptibility in DM patients, as compared to the controls, which was, to the best of our knowledge, the highest reported. The salient results, reconfigured for comparison, are depicted in Figure 19.

The bar diagram above depicts the differences between diabetic (black bar) and non-diabetic controls (gray bar). Susceptibility to COVID-19 and higher BMIs were evident in those with diabetes. Many other parameters were also statistically significant in the same direction, except for alcohol consumption, which was higher in the controls.

Since our last data-driven topic had involved the role of diabetes, we decided that a cursory look at the effects of commonly used anti-hypertensive medications on COVID-19 deserved summarization, as depicted in Table 5.

## 3. Discussion

The data described could be indicative in defining susceptibility to COVID-19 using FERAD scores and is awaiting confirmation in a larger dataset, but at the current juncture, it did appear to predict severity. The actual COVID-19 sample number was small, particularly when considering that we had FERAD scores for only 35% of patients. This could be further refined, as of the 2 components in the ratio, only p87 had been closely dated to the time of enrollment, while the dates of the ferritin levels were clinically determined by the primary provider. In addition, many of our participants never had a ferritin draw. Despite some cases of iron deficiency or iron overload [36], the body appeared to maintain its ferritin levels within strict limits, in a non-end-stage renal disease situation. In the event of multiple ferritin readings, if the time differed from the stool sample, it could be beneficial to obtain a composite average.

To expand, although our surrogate model may not be directly applicable to the general population, it appeared to be relevant to the AA population, particularly those who have a FERAD ratio. The COVID-19-positive group was small, but it represented the many problems and complications that could be expected in this disease [3] and provided us with the rationale of using FERAD ratios as the significant differential features between the diseased patients and the controls. It appeared that the immune system was dependent on the intracellular iron stores for optimal functioning and closely regulated these resources under usual conditions. The past and current ferritin levels correlated in our predominantly male patient sample, suggesting that there could be an interaction between the iron hemostasis reflected by ferritin levels and the innate immune system. This association, when stool p87 levels were low, resulted in a high FERAD ratio, which correlated with the low overall survival within a pre-COVID-19 surrogate population. Prospective studies should ascertain patients with high ratio levels and randomize these to form a group who will receive stringent protective measures and be allocated all available anti-viral resources. The outcomes of this group should be contrasted to those from a group with low FERAD ratios who would be treated with the current standard measures.

The question of which factors make this disease a unique threat to the AA community could be addressed by this study. Twelve percent of the AA population carried a mutation in the Jak3 gene that could be associated with reduced expression of p87 in the proximal colon, which is the typical region for the Paneth cell distribution [37,38]. We observed some reduction in the p87 expression, which was limited to a single right and a single left colonic region in this study. This suggested that a mutation could have led to a diminution of Paneth cell numbers or functions with the lower production of stool p87, resulting in a low FERAD ratio and a worse outcome. High stool p87 ELISA results could be associated with this disease, as observed in our small series of COVID-19 patients. This could also explain why the only significant differences were observed in AAs when the surrogate modeling was applied to historic FERAD ratios. The significant differences when transitioning from a higher-ratio quartile to a lower, with different survival outcomes, supported the differential effects of the ratio on survival not observed in the Caucasian patients. We examined some, but not all, of the commonly accepted adverse risk factors [3] in this study.

Paneth cell metaplasia has also been shown in the pancreas, uterine cervix, ovary, and prostate tissues, confirming relevance to other organs where p87 had been produced and, consequently, may have had a diagnostic or prognostic role [39]. Therefore, if the innate immune system effector cells, such as Paneth cells, were the key to both survival and florid, adverse outcomes, and if p87 is proven to confer resistance to infection, survival advantage, and a treatment option, it may also be useful to determine the effects of medications on p87 expression. These could influence the form of stool p87 produced, which could be advantageous in blocking viral effects and spread. This could serve as a potential adjunct to the relatively few therapeutic agents available to clinicians in the current battle being waged in the face of a relative dearth of positive information regarding the potential medical armamentarium needed to contain COVID-19.

Similar to the FERAD ratio, we found attributes for prognoses and severity in the absolute neutrophil/absolute lymphocyte ratio that appeared to reflect a contribution from both the innate immune system effectors (neutrophils) and, perhaps, a mix of innate and adaptive immune systems in the lymphocyte component. Interestingly, this ratio was able to predict many cancers, including esophageal, breast, colorectal, prostate, and lung [40], many of which had also been predicted using stool p87 results. In an accompanying editorial to reference [41], the writer asked a question germane to the roles of both NLR and FERAD: ”Can a single NLR value from 3 months before index endoscopy truly predict Barrett’s Esophagus progression several years later?” We believe so, and the present data showed that there was a significant positive correlation between the NLR values taken at the time of stool collection and then later at the time of the COVID-19 infection, in our COVID-19 patients. It was important, however, to focus on the larger, prospective COVID-19 studies using this biomarker, as we had limited data, which made extrapolations difficult.

While the clinical epicenter of the morbidity and mortality of COVID-19 is the lung, and, by physiological association, with the heart, it appeared that the GIT did not lag far behind. The lung is limited in its defensive capabilities, as its major function of gas exchange is reliant on the thickness of the respiratory alveoli. The lung richly expresses the protease ACE2, which then yields angiotensin 1–7 to activate many beneficial activities via the MAS receptor in the lung vasodilation and the suppression of inflammation and fibrosis [42]. These authors pointed out other beneficiaries, such as the kidneys and the heart, but cautioned that the ACE2+ macrophages carrying the virus might be attracted to the alveoli and spread the infection, whereas the dendritic cells, resistant to viral replication, would not. They asserted evidence that correlated activated lung macrophages to the severity of the disease. This led us to consider the state of the alimentary canal, as compared to proliferation. In their otherwise thoughtful and timely review of the human InImS, via COVID-19, Schultze and Aschenbrenner failed to address the role of the Paneth cells [43]. They importantly drew attention to NRP1, which is an important viral cell-penetrant, particularly in cells with a low ACE2 expression. This could be an ideal description for Paneth cells [18], and the authors indicated that immune cells did not appear to be a target of the SARS-CoV-2 virus. Therefore, as previously mentioned, Paneth cells, in addition to expressing p87 and being resistant to COVID-19, secrete aDef5 that avidly binds to ACE2 at 73.2 nM while the SARS-CoV-2 virus binds at 14.7 nM [44]. High concentrations of aDef5 [45] have been found in the ileal fluid (6–30 µg/mL), making it unlikely that the SARS-CoV-2 virus could displace the avid binding of aDef5 found in abundance in the bowel. It was then difficult to accept that the reported intestinal replication and fecal shedding of the virus occurred after a relatively long period after nasopharyngeal detection and, at times, preceded it. It appeared untenable that there should be an unopposed activation of aDef5 in much of the bowel, as ACE was highly expressed in the enterocytes and the colonocytes, as well as in the liver and biliary tract [46]. Clearly, there is much to be learned about defensins and COVID-19. We have shown an association of aDef5 and p87 in the pancreas [18] but not in the lung, since aDef5 did not appear to be normally expressed in normal mucosa in this organ. Beta-defensins have, however, been expressed in the lung, and their role should be explored [47]. The use of aDef5 as a therapeutic measure in the lungs would be, at this juncture, ill-advised.

The LAI test mediated by OSN [48] was likely an innate immune system phenomenon, unrelated to p87, and accordingly, we did not find any correlations in the p87 testing. There appeared to be correlations between PSA and urinary creatinine that may all be common factors when using that medium [39]. Of some interest was that the effluent and the stool p87 did not show any significant positive correlations. They were different in how and when they were collected, and the higher levels in the effluent could have reflected those differences.

Females appeared to have better outcomes with COVID-19, as compared to males, in general [49], but our modeling was dominated by the significant age differences, potentially explaining the differences, but the lower FERAD ratios suggested lower susceptibility and, conceivably, better outcomes. There were no females represented in the highest ferritin/stool87 quartile 4, which had the highest mortality.

A recent study [49] showed it was possible to develop a computer model for prediction that incorporated significant factors, such as AA ethnicity, males, and the elderly. Those at less risk were those who had influenza and pneumococcal vaccination. Those taking medications, such as melatonin, paroxetine, and carvedilol, were also at reduced risk. This model was not applicable to the individualized FERAD scores that we described. However, we confirmed a role for other proven biomarkers in COVID-19 [44,50] and showed that the FERAD ratio could also serve as both a susceptibility and risk marker for serious disease, as well as a very useful function for prognoses. The ability to use different medications to achieve an optimal FERAD ratio could be part of a preventative strategy to contain COVID-19 and its variants.

Our data suggested that a Paneth-cell immune modulator marker was reduced in the stool of patients taking ACE-I/ARB, thereby reflecting a modest deficit in the innate immune system. Whether this influenced the response to COVID-19 remains unknown. CCB treatment, which has antiviral activity [33,34], could be a suitable alternative for some, but not all, conditions for which these agents were prescribed. The benefit-to-harm ratio of continuing to take ACE-I or ARB, which are effective in the treatment and prevention of substantial cardiovascular, diabetic, and renal morbidity and mortality, should be addressed by a comprehensive epidemiological study utilizing reliable available data. We would welcome a comprehensive international database that includes co-morbidities, medications, and clinical outcomes that could provide insights into these issues and guidance for patients and practitioners. Until such a time as these data are available, we cannot endorse changing prescribing recommendations.

Several publications have expressed concern regarding using these medications, but in 2021, with the publication of an extensive meta-analysis [35] of COVID-19 patients from the BRACE-CORONA and the REPLACE-COVID studies, which concluded that hospitalized patients who had continued these medications had no additional deaths, as compared to patients who desisted. The meta-analysis included 101,949 patients and 16,545 patients taking ACEi/ARB medications drawn from 40 international studies, including from China and the West. The data were offered as adjusted and unadjusted, but there were no additional deaths in the patients taking these medications for the assigned indications in these studies, as previously discussed. However, in the adjusted dataset, there was a reduced chance of dying if patients continued taking their medications (OR 0.57 (CI 0.43–0.76)), both in patients with hypertension and a plethora of co-morbidity. This practice is now widely supported by professional societies internationally.

In terms of the stance of the neonatal InImS that showed a correlation between Src and p87, but not Abl2, the latter is an important kinase for cell development, including neuro-synapse transmission by phosphorylation, but Abl2 can often be redirected by pathogens to enable the benefits of host-cell actin cytoskeletal changes and enhance virion egress from the cell. In COVID-19, the inhibition of Abl2 by imatinib has been thought to block the Abl2 effects of VEGF on the pulmonary endothelium, a common ARDS scenario, as observed in COVID-19 [51]. Since Src is important in signaling, its membrane-bound form, when activated by ATPA1 (which encodes Na^+^, K^+^-ATPase), has been shown to inhibit both coronavirus and respiratory syncytial viral entry, leading to the consideration of using specific inhibitors, such as ouabain and digoxin, which have relatively better toxicity safety profiles, to attenuate COVID-19 [52]. It was of interest that p87 and Src were closely correlated and could share certain functions. MAP kinase p38 and its downstream effectors were instrumental in the release of IFN-1 in response to viral pathogens [53]. It was also clear that the robust expression of IFN and other immune signaling genes in children, as compared to adults, could account for milder disease [54], with the exception of multisystem inflammatory syndrome in children (MIS-C). We postulated that these detected moieties, if extant at the time of infection, could have somewhat attenuated COVID-19 severity. This somewhat different InImS stance in children with viral disease appeared to be a paradox and is not well understood, despite intensive study, but relatively little data from the COVID-19 era is available. The immune systems of adults and children appeared to differ, and the presence of multiple pathogens simultaneously presented in children, particularly in the lungs, could modify the response. Differences in the Toll-like receptors (TLR) and macrophages likely play a role, but the neonatal case report presented, regarding the likely Paneth-cell stool response, could lend weight to the concept of “immune-rage at an earlier age”.

There are three stages of responding to an epidemic: prevention, containment, and allocation of resources for effective treatment. However, the actual approaches can vary, and for every location, there must be adaptation. Avoiding exposure is logical, but the extremely contagious nature of SARS-CoV-2 and its emergent variants makes this difficult outside of a BSL2 laboratory setting, but it certainly can be effective, as observed in the initial outbreak in Wuhan, China. What cannot be changed and may harm affected individual outcomes are the factors of age and the often-related underlying disease states. Much information is needed as to what constitutes an underlying disease. Are patients with well-controlled underlying disease equally at risk from harms as those who with uncontrolled disease? Does the alteration of the immune response, either from immunodeficiencies or with something innocuous as a past tonsillectomy or appendicectomy, place a patient at elevated risk in general? At this point, this does not appear to be the case (unpublished data for tonsillectomy effect could result in being predisposed to chronic fatigue). Why are we seeing different results in patients with similar severity when using the same anti-IL-6 tocilizumab in patients with severe COVID-19 pneumonia [55] and those starting ICU life support with 2 additional arms but both with placebo controls [56]? Paradoxical responses in the immune system have been described and may provide an explanation [30]. It would appear that commonplace medications administered or taken regularly by patients at risk should also be scrutinized for benefit or harm, but as described previously, there were numerous candidates that could have been used based on reliable research data already published.

This study suggested that defining a threshold of FERAD in prospective COVID-19 cases, using the mean-plus-two standard deviations observed in group 1 (22,461) or determining the presence of specific p87 bands in the stool, could enable triage and early treatment, for example, with the FDA-approval of anti-virals, molnupiravir, PF-07321332; paxlovid; ritonavir combination, from the highly nuanced Phase 2/3 EPIC-HR Study (https://www.pfizer.com/news/press-release/press-release-detail/pfizers-novel-covid-19-oral-antiviral-treatment-candidate, last accessed on 15 December 2022); and the recently published fluvoxamine TOGETHER treatment study (NCT04727424), showing significant reduction in relative risk of death with the medication, current ongoing in 11 centers in Brazil [57]. Low-cost options for treatment are ASA [58]; experimental ACE2 blockers that have completed the animal stage of experimentation [59], and iron deprivation/chelation are further promising strategies in the battle against the virus [60,61] since MHC expression and NK cell recognition are affected by iron [62]. Other potential targets are RNA-dependent RNA polymerase [63] and high-dose dietary supplements [64].

FERAD could thus define at-risk populations and enable time-based therapeutic interventions to reduce poor outcomes through use of the early intervention medications, once approved. Specific p87 bands in AAs could confer a better overall survival rate. Since stool p87 originated in Paneth-like cells, intact p87 bands could have reflected the optimal activity of the InImS, which could allow for stratification and triage. Of note, p87 has also been found in the lungs in cancer and other inflammatory conditions [65], the epicenter of mortality, especially relevant to the current pandemic. Interestingly, p87 also has some genetic homology with an important lung chloride channel protein hClCa-1 [66] that activates pulmonary macrophages. AAs, an ethnic group with worse outcomes after contracting the virus, had FERAD ratios of 46,537 ± 86,452 in the group 1 of COVID-19 patients, and diabetes prevalence was equivalent in both AAs and Caucasians. There were no differences in the adenoma detection rate at the index colonoscopy (~67%) in those patients who underwent an index colonoscopy and submitted stool specimens (47% of 1081 AAs and 41% of 964 Caucasians).

## 4. Materials and Methods

### 4.1. Sites and Participants

The demographic data, endoscopy, pathology, survival, medication history, ferritin results, and COVID-19 status were obtained from patients deemed to be at increased risk for colorectal neoplasia by a standardized patient questionnaire and the data retrieved from the Veterans’ Health Administration (VHA) CPRS. Since there may be sex differences in viral response, we collected data from both sexes. In order to better define the origin, shedding, and systemic circulation of the antigen, a subset of patients who submitted stool samples were offered to participate in a more detailed collection (Phase 2 of the CRC study) consisting of 3 cold forceps biopsies and colonic washing aspiration (effluent) of their own volition. Colonoscopic biopsies were obtained during a colonoscopy at 6 regional sites of the colon (cecum, ascending, transverse, descending, sigmoid colon, and rectum) by cold biopsy forceps, 1 biopsy sample was used for immunohistochemistry (using the above alkaline phosphatase Vector Laboratories kit, but using EAC (aminoethyl carbazole) as the substrate), as previously published [18]. Immunohistochemistry results were graded semi-quantitatively, as described previously [25]. To investigate potential contributions of the other components of the InImS [15], we also used a monoclonal directly against urinary organ-specific antigen (OSN) standardized by urinary creatinine in an ELISA. OSN is the antigen that mediates the lymphocyte adherence inhibition test (LAI), and we used the OSN urinary test that we had previously found to be useful in CRC diagnosis [48]. The blood-based LAI had also been useful in the diagnosis of cancer of the urological tract [39] and was likely part of the innate immune system, which enabled comparison and finding associations with other elements of the InImS.

Informed consent was obtained from all participants, with Phase 2 patients signing a second consent for the tissue biopsies. The studies were approved by the Wayne State University School of Medicine Internal Review Board, and written informed consent was obtained.

### 4.2. Chemicals and Antibodies

#### 4.2.1. Stool Extraction and Antibodies

The stools were extracted from cards that were cut out and vortexed in phosphate buffered saline (PBS), and protein content was determined by use of the Bradford method (Bio-Rad, Hercules CA, USA). The p87 antigen was defined by the monoclonal antibody Adnab-9 and compared against adenoma antigens by the principal author; it was an IgG2a-isotype grown in mouse ascites. Its origin and neoplasia biomarker capacity has been published [18,25].

#### 4.2.2. ELISA, Western Blotting and Chemicals

The ELISA was standardized for protein (5 µg/well), and Western blotting for specific bands was performed, as well as in the subset of patients who signed an additional Phase-2 consent for regional colonic biopsies, as described above, and subsequent immunohistochemical Adnab-9 labeling. The ELISA assay and Western blotting had been described in detail and published previously [11]. Briefly, microplate wells were instilled in duplicate wells with the stool extracts at 5 µg protein/50 μL PBS per well Primary Adnab-9 (previously available from DakoCytomation, Carpinteria, CA, USA). Anti-p87 monoclonal murine ascites was applied at a 1:500 dilution followed by secondary goat anti-mouse biotinylated antibody, followed by an avidin linker with alkaline phosphatase from a commercial kit (Vector Labs Inc., Burlingame, CA, USA) as part of a standardized kit and performed according to the manufacturer’s protocol. Antibody-bound antigen was detected with a pNPP (p-nitrophenyl phosphate solution—Sigma, Santa Fe, NM, USA), and the wells were read with a microplate reader at 405 nm wavelength (ThermoFisher Scientific Waltham, MA, USA). Control wells were treated with a primary anti-levan (plant carbohydrate) murine monoclonal antibody (UPC 10 Sigma Aldrich, St. Louis, MO, USA) of the same Ig2a isotype as Adnab-9. The assay reproducibility was highly specific even after a decade of −70 °C storage [15]. The data were expressed as OD-background/5µg protein. For the purposes of calculating the FERAD ratio, if the ELISA result was zero, we regarded it as 0.001 to be able to include these patients. Western blotting was likewise achieved using components of an immunoperoxidase kit using 3,3′,5,5′-tetramethylbenzidine (TMB) as substrate (ThermoFisher Scientific) using p87 isolated from pooled colorectal adenoma extract as the positive control in each procedure. Ten µg of protein sample were electrophoresed on a 9.6% sodium dodecyl sulfate-polyacrylamide gel and then transferred to PVDF membrane, after which Adnab-9 was applied in the ABC technique, as described above, to reveal specific bands, as compared to molecular weights standards run simultaneously to allow for determination of the relative mobility of the detected bands. An identical membrane was processed with an irrelevant monoclonal of the same isotype as Adnab-9 to ensure specificity. In the course of the characterization of the Adnab-9 monoclonal antibody, stool was collected from a healthy infant for 2 months. Amongst other markers, p87 binding was performed in the ELISA, as described above. Fortuitously, binding of other available antibodies now having COVID-19 relevance, *Src* and *Abl*, (antibodies were gifted by Dr. A. Majumdar), were also able to be performed. As described above, p87 Western blotting and ELISA of the adult and neonatal stool (obtained from a previous study) were performed.

### 4.3. Statistical Analysis

Parametric (Student’s *t*-test) and non-parametric χ^2^ testing values for differences in proportions to yield odds ratios and confidence intervals were used for the analysis of continuous (ordinal) and non-continuous (non-ordinal) data. Linear correlation analysis by the difference of the least-squares method was used. All *p*-values less than 0.05 were considered as significant associations, but no causality was necessarily implied. The online “VassarStats” statistical program was used intermittently from 2020 to March 2023, (http://vassarstats.net/) for statistical analysis.

## 5. Limitations

With respect to limitations and strengths, our COVID-19 patient numbers were small, and there was also a risk for type-2 errors in at least 5% of our calculations. However, we had fairly large numbers of stratified controls to ensure that our comparisons were valid. We were unable to provide a cost estimation for many of the tests performed, such as the most basic NLR up to the most expensive PCR and Western blotting analyses. NLR testing also had its limitations and may not have been applicable for untoward outcomes of COVID-19, such as long COVID, and this also applied to Adnab-9 p87 stool assays, which have not been commercialized. We are also unsure whether our data can be generalized to the non-AA population.

Our strengths were the long historic follow-up data we had for most of our patients; the prospective nature of our study that reduced bias; and the self-selection of patients in an entirely voluntary study with data collection prior to the COVID-19 pandemic. We emulated the work of a remarkable team that showed the validity of being able to reach firm clinical conclusions based on the expeditious study of relatively small groups of patients [67]. We also performed multiple reproducibility studies on our frozen, stored samples and found that the reactivity had been well conserved over 10 years and was highly reproducible [68]. We also showed the robust nature of the antigen, as the bands were detected after 6 days at room temperature. We look forward to expanded studies to confirm our findings and invite collaboration, particularly in the promising area of the gut microbiome [69,70].

## 6. Conclusions

The COVID-19 pandemic presents an extremely complex set of issues that have been likened to Pandora’s box in Greek mythology (the Pandora pandemic) and has spawned its own medical myths, raising important questions concerning trust and belief. In this manuscript, we attempted to grapple with multiple issues and render them more understandable. We believe that the potential ability to gauge the stance of the InImS may be a novel finding. Its effect on both disease susceptibility and outcome, in both children and adults, including the effects of underlying disease, is intriguing. We believe this may represent a shared accomplishment, as we acknowledge those researchers that have provided advances upon which we could build. We sought to be inclusive and introduced potential prognostic biomarkers developed by others, as they related to our patient population. This was important because it is only with global cooperation and goodwill that humankind will overcome this viral scourge, and the others yet to come. In this vein, we posed critical questions that demand answers that hopefully will come from international collaboration. In this vein, we recently highlighted other patient groups at risk of the COVID-19 pandemic [71], adding to what Charles E. Rosenberg in 1989 coined as “dramaturgy” [72]. May the current scientists prevail.

## Figures and Tables

**Figure 1 ijms-24-07536-f001:**
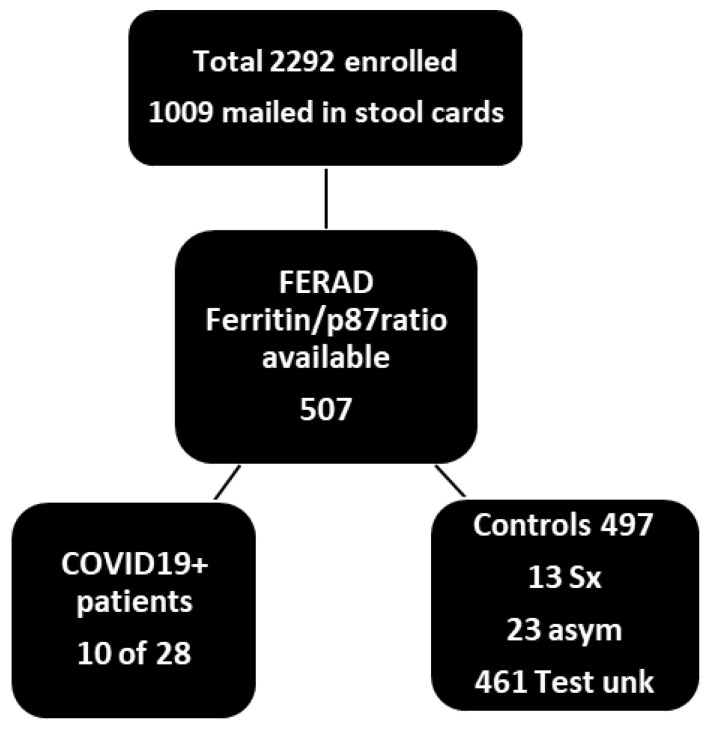
Flow diagram of patients who submitted stool cards and had ferritin levels for a FERAD calculation. Less than 50% of patients turned in cards, and only 507 of those had ferritin concentration levels determined opportunistically and available from the computerized patient record system (CPRS). FERAD ratios in symptomatic (Sx) and asymptomatic (asym) patients totaled 36 and 461, where the results of a PCR COVID-19 test were unknown (unk). The proportion of stool cards delivered by postal mail was not substantially different from the known return proportion of 44% for occult blood card return, as determined for that time period at the Detroit VAMC.

**Figure 2 ijms-24-07536-f002:**
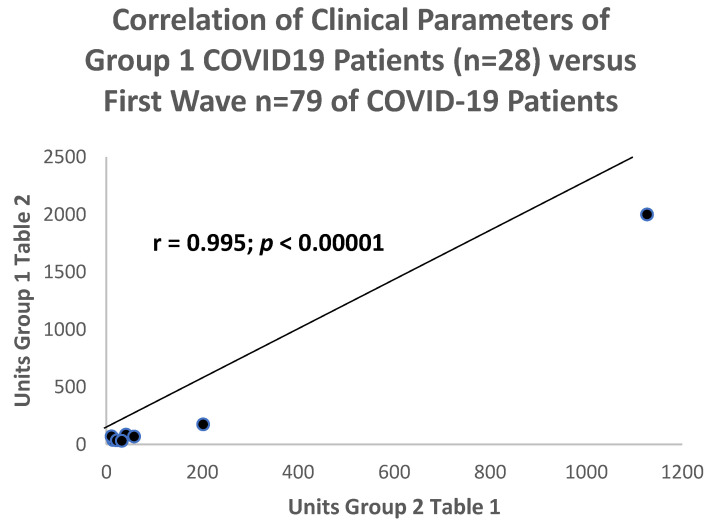
Scattergram of comparable clinical parameters in the current COVID-19 cumulative cohort versus the same parameters in the first wave of COVID-19 patients from the same medical center.

**Figure 3 ijms-24-07536-f003:**
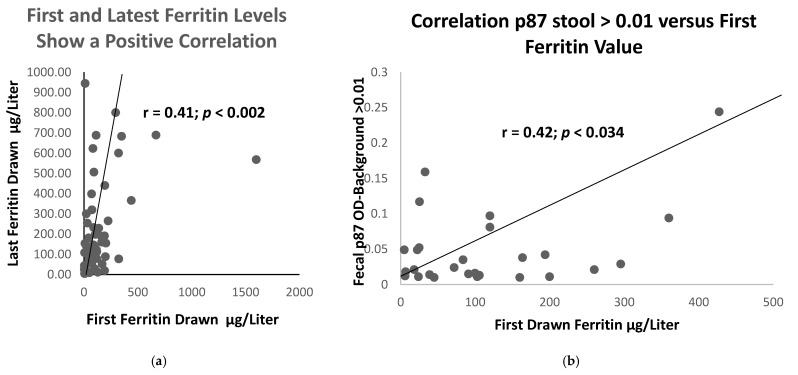
Correlation of paired ferritin levels, initial and last drawn in 56 patients (**a**), and a scattergram of 27 pairs (**b**) of initial ferritin, plotted against Stool bound Adnab-9 binding > 0.01 showing a positive correlation between stool p87 and initial ferritin values.

**Figure 4 ijms-24-07536-f004:**
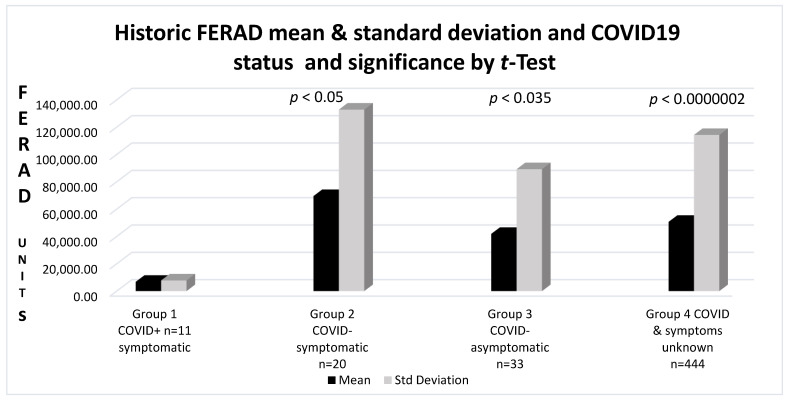
A bar diagram depicting pre-morbid FERAD values of COVID-19-positive patients, as compared to controls and COVID-19-negative test patients.

**Figure 5 ijms-24-07536-f005:**
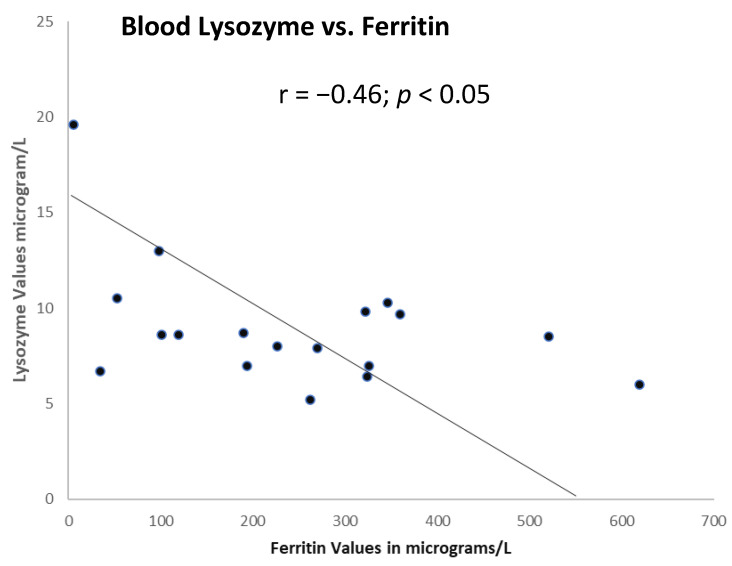
Significant correlation between lysozyme and ferritin in the serum.

**Figure 6 ijms-24-07536-f006:**
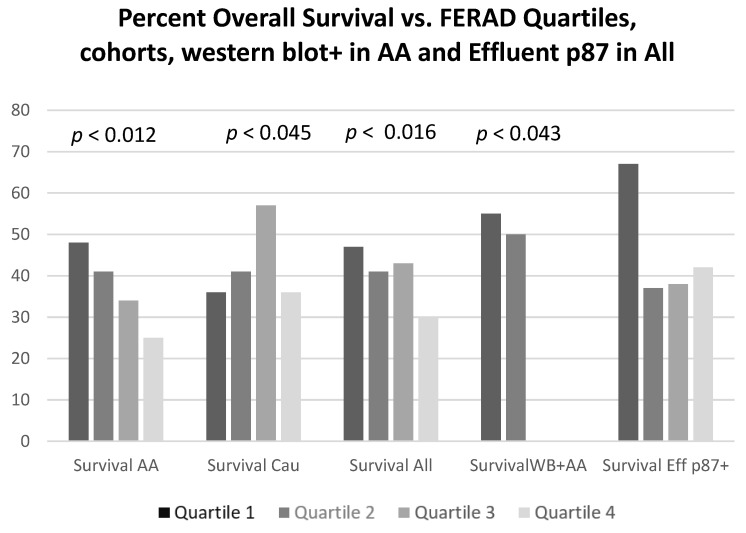
Bar diagram of overall survival by FERAD ratio quartile. Abbreviations: AA—African American; Cau—Caucasian; Eff—Effluent; WB—Western blot.

**Figure 7 ijms-24-07536-f007:**
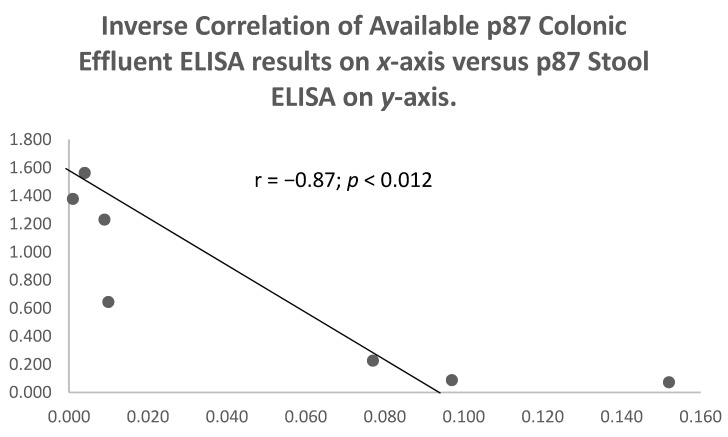
Scattergram of an inverse correlation of p87 stool bands and overall survival in Group 2 patients.

**Figure 8 ijms-24-07536-f008:**
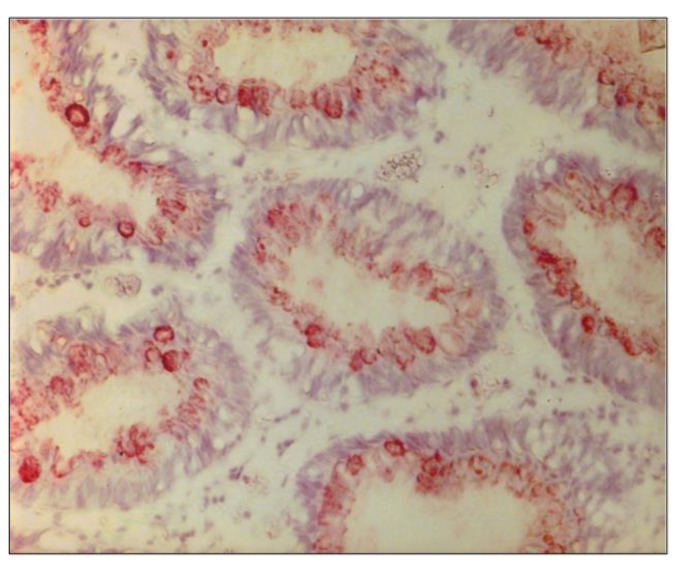
Photomicrograph ofAdnab-9 staining by immunohistochemistry in normal-appearing colonic mucosa. This photomicrograph with a particular magnification of 50×, in a normal colon biopsy section, showed the strong specific reddish-brown staining in the cytoplasm.

**Figure 9 ijms-24-07536-f009:**
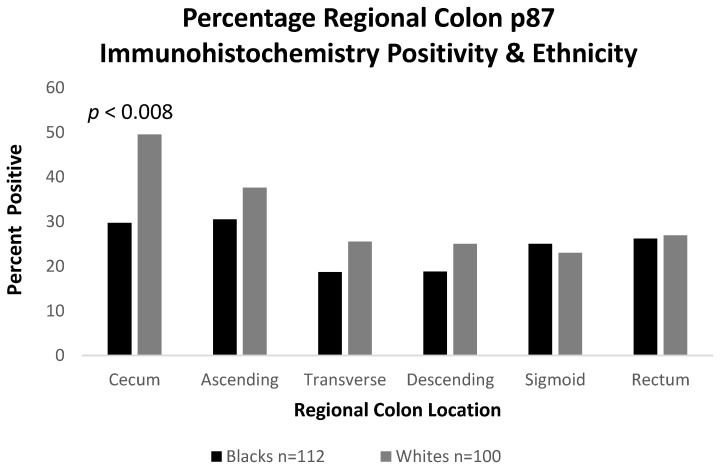
Bar Diagram showing reduced fixed p87 antigen in the ceca of AA, as compared to Caucasians.

**Figure 10 ijms-24-07536-f010:**
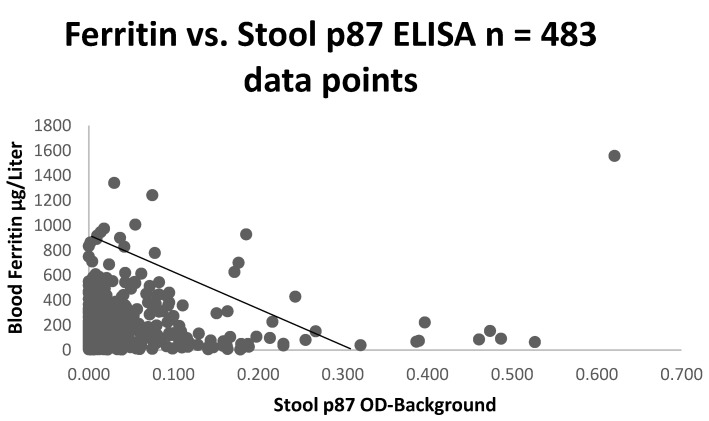
Scattergram of stool p87 negatively correlated with ferritin.

**Figure 11 ijms-24-07536-f011:**
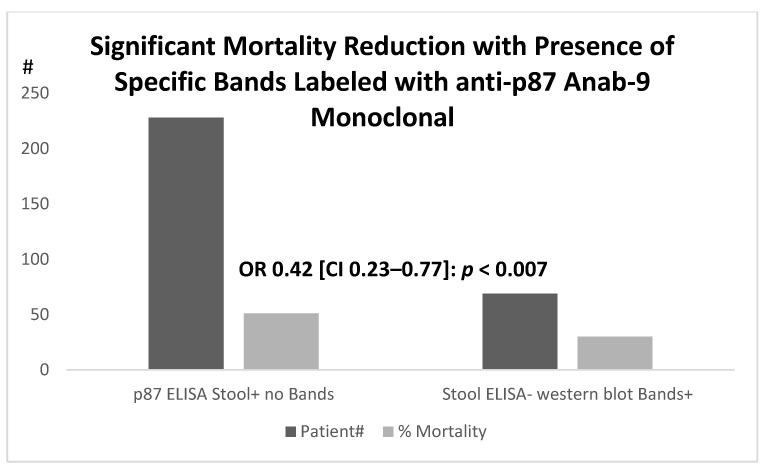
Relationship of outcomes in patients with positive fecal p87 ELISA (p87 ELISA stool+) and those with negative stool ELISA but detectable p87 bands in stool (Bands+).

**Figure 12 ijms-24-07536-f012:**
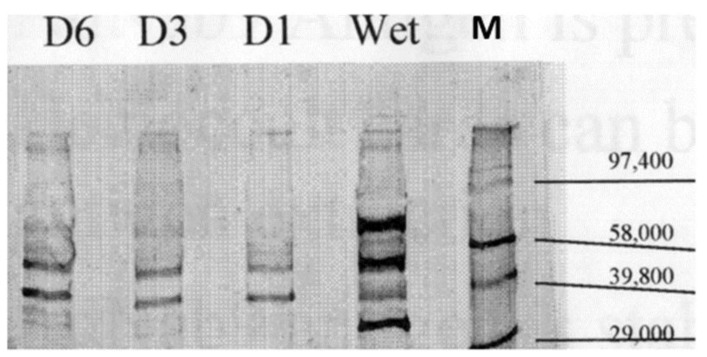
Western blot of stool with specific p87 bands and Molecular Weight Markers. Monoclonal antibody Adnab-9 that recognized p87 reacted in a Western blot of fecal samples. The blot showed positive bands with the relative mobility (M) molecular weight standards designated on the right. “Wet” represented a sample that had been freshly obtained and processed less than 24 h, being left at room temperature. D1, D3, and D6 were the sample stored at room temperatures for 1, 3, and 6 days, respectively. Paradoxically, while D6 appeared to have stronger bands, as compared to D1 and D3, the conditions of prolonged changes in the temperature and humidity could have caused these changes in the protein bands.

**Figure 13 ijms-24-07536-f013:**
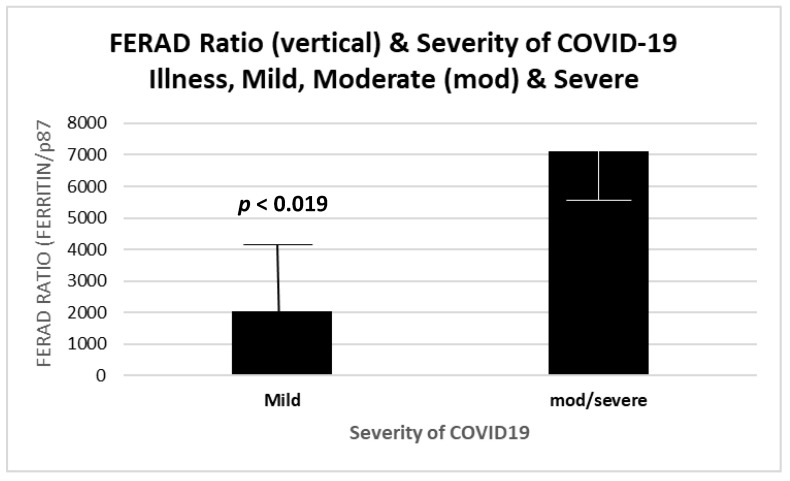
Mean FERAD ratios increased with the severity of COVID-19.

**Figure 14 ijms-24-07536-f014:**
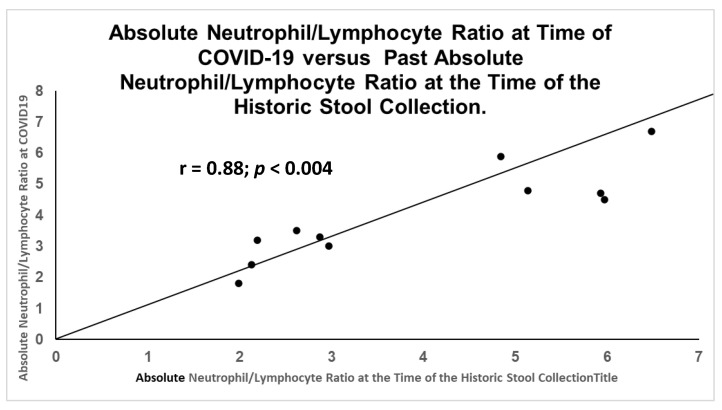
Direct significant correlation between sets of NLR obtained at time of initial stool collection and COVID-19.

**Figure 15 ijms-24-07536-f015:**
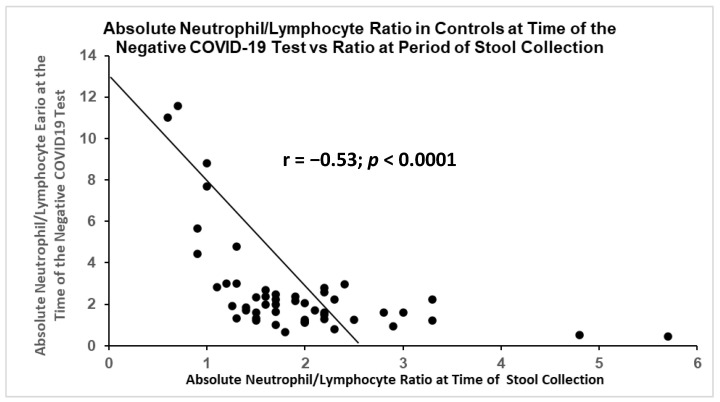
Inverse correlation between sets of NLR obtained at time of initial stool collection and during time of non-COVID-19 related symptoms.

**Figure 16 ijms-24-07536-f016:**
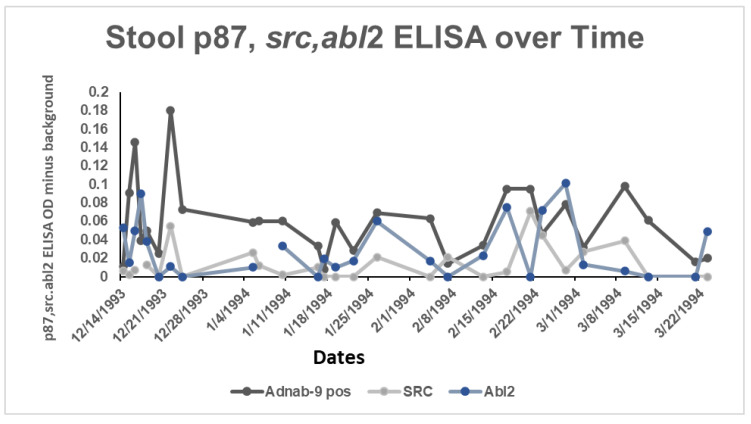
Depiction of 3 biomarkers expressed in the stools of a neonate during the first months of life as determined by ELISA.

**Figure 17 ijms-24-07536-f017:**
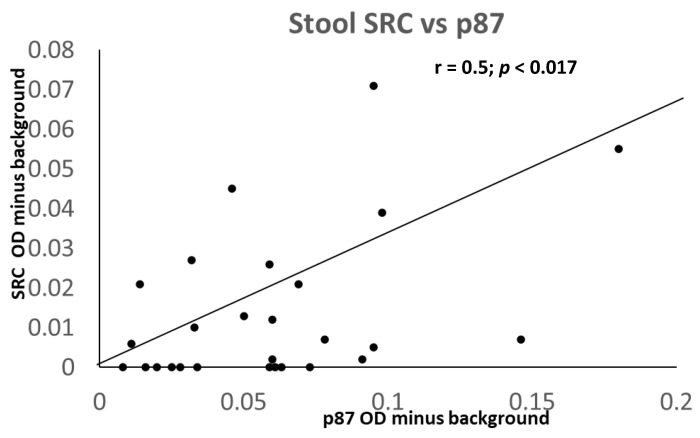
A Scattergram showing a positive correlation of Src and Adnab-9-defined p87 in the infant stool.

**Figure 18 ijms-24-07536-f018:**
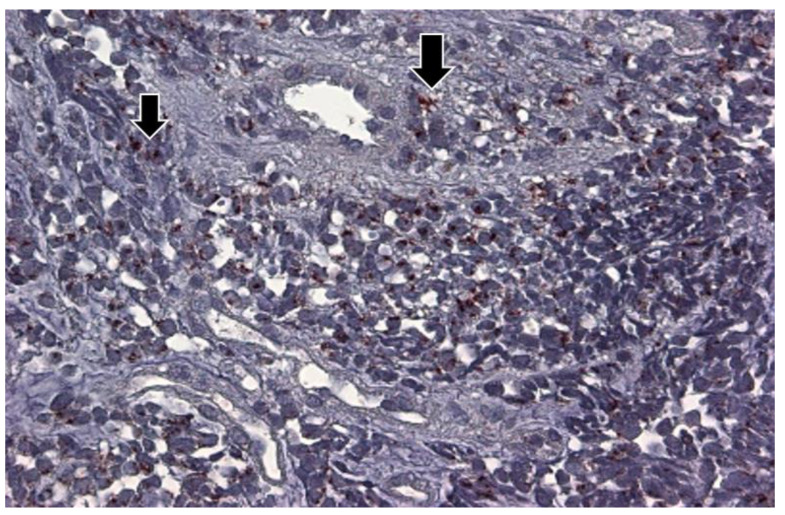
The p87 immunohistochemical labeling of lung adenocarcinoma tissue at intermediate power of magnification (25×). The photomicrograph shows the adenocarcinoma diffusely involving the lung, but some normal bronchioles were observed (arrows) that exhibited a deep reddish-brown substrate, indicating the presence of p87. The dark brown p87 was observed in the cytoplasm of clearly malignant adenocarcinoma, shown on the left (smaller arrow), and in normal cells located in the upper center (large arrow).

**Figure 19 ijms-24-07536-f019:**
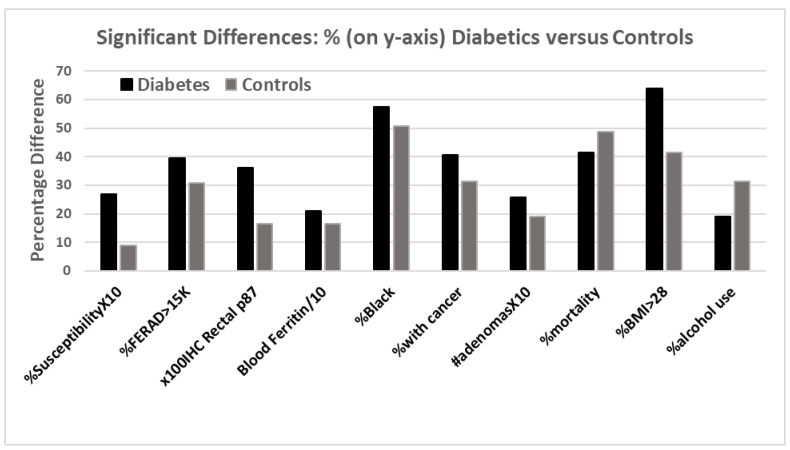
A bar diagram showing significant differences between the patients with DM and controls.

**Table 1 ijms-24-07536-t001:** Demographics and hospitalization data in COVID-19 and control groups.

Demographic n = 2292	Group 1	Group 2	Group 3	Group 4
PCR/Symptom Status	COVID-19 + ve/Sx + ve	COVID-19 − ve/Sx+	COVID-19 − ve/Sx − ve	PCR/Sx Unknown
Number (percent)	28 (1.2)	36 (2)	90 (3.9)	2129 (92.9) ^
Entry/testing (Age ± SD)	54 ± 11/72 ± 9.7 years	54 ± 8/73 ± 8.6 years	54 ± 8.6/73 ± 9.0 years	61 ± 12.4 */NA years
Sex (%male = m) NSS	m = 22 female = 5 (81.5)	m = 36 female = 9 (80)	m = 72 female = 13 (84.7)	m = 1684 female = 209 (89)
Ethnicity (%AA):White	17 (63):10 (37)	33 (73):12 (27)	54 (60):36 (40)	96 (51):969 (49)
Mortality number (%)	5 (18.5)	5 (10.9)	5 (5.6) *p* = 0.05 vs. 1	NA
COVID-19 test (%)	22 (82%) known	41 (89%) known	49 (53%) **	NA
Vaccinated (%)	yes 15 (56%) no 12 (44%)	yes 22 (54%) no 19 (46%)	yes 40 (66%) no 21 (34%)	NA

Sx + ve–symptomatic; ^ *p* < 0.0001 group 4 vs. group 1 by ANOVA and Dunnett’s post hoc analysis; * *p* < 0.0001 as compared to Group 1 by *t*-test; NSS—not statistically significant; AA—African American; NA—not available. Diagnostic tests were SARS-CoV2Xpert Xpress assay, Cepheid GeneXpert instrument, starting 30 March 2020; Fluvid (also tests for influenza A&B later RSV), PHRL (none in Group 3), PCR outside hospital, where test was unknown, it was not included in the data.** *p* < 0.014 vs. Group 1, and *p* < 0.0001 vs. Group 2. All vaccine recipients received at least 2 doses of mRNA-1273/Moderna vaccine.

**Table 2 ijms-24-07536-t002:** Clinical details of PCR confirmed group 1 COVID-19 patient and first-wave patients with comparison to PCR-negative controls with signs of infection.

Clinical Parameters	Group 1 COVID-19 Patients (Data%) cf Denotes First Wave	Group 2 Non-COVID-19 Controls (%)	*p*-Value *
Severity of disease	14 no signs/mild; 12 severe (96)	29 mild 15 severe (96)	NSS
Days of hospital stay	5.81 ± 8.66 cf 6 (4–11.5)	1.09 ± 4.4	**<0.02**
ICCU admission	4 yes 23 no 14.8% cf 37.1%	0% (OR 7.83 [0.83–76.11])	**=0.017**
Pulse oximetry %	94.2 ± 5.2	96.1 ± 2.2 *p* = 0.15	0.15
Hemoglobin 12.5–15 d/dL	12.2 ± 2.6	12.5 ± 2.5 *p* = 0.67	0.67
D-dimer 200–250 ng/dL	1993 ± 2683 cf median 508	1145 ± 1435	0.22
CRP	10 yes 0 no (100%)	8 yes 14 no (36) (RR 2.75 [1.58–9.78])	**<0.002**
Ferritin ng/mL	1127 ± 1598 cf 2000	167 ± 142	0.062
Platelets × 10^7^/L	202,046 ± 56,591 cf 173,000	226,222 ± 85,853	0.24
Absolute monocytes × 10^7^/L	0.734 ± 0.892	0.684 ± 0.330	0.71
Absolute lymphocyte × 10^7^/L	1.224 ± 0.646 cf 0.9	1.646 ± 0.633	**=0.014**
Ratio abs m/abs ly	1.005 ± 1.716	0.473 ± 0.374	0.17
Ratio abs ly/abs m	2.305 ± 1.179	2.863 ± 1.478	0.13
LDH	391 ± 142	355 ± 162	0.60
Supplemental oxygen	11 yes 16 no (41%) cf 83.5%	2 yes 43 no (4%) OR 14.78 (2.95–74.12)	* **<0.0002** *
Remdesivir/chloroquine	3 of 27 each (11% each) cf 69.6%	N/A	N/A
Steroids	6 yes 21 no (22%) cf 34.2%	1 yes 44 no OR 12.86 (1.45–113.68)	* **<0.009** *
NSAIDs	14 yes 13 no (56%)	25 yes 20 no (56%)	NS
BMI (kg/square meter height)	32.5 ± 5.823 cf 30.7 ± 7.6	30.1 ± 5.4	0.20
Diabetes mellitus type 2	15 yes 11 no (58%) cf 60.8%	22 yes 24 no (48%)	0.47

Incorporates symptoms, oxygen use, CXR findings, and mortality. * Statistically significant values shown in bold. Italicized *p*-value data are shown from a descriptive paper of 79 patients from the first COVID-19 wave, 10 March 2020 to 6 April 2020, at the same institution for comparison (cf), where applicable [17]. Our patients were older with a mean 72 versus 69 years; despite the differences, the mortality was higher, 18.5 versus 25.3%, respectively, but this was at a point in time when treatment modalities were at a preliminary juncture with higher comorbidities/inflammatory markers. There may be overlap in the patient groups, as we had no access to patient identities in that study. NSS—not statistically significant.

**Table 3 ijms-24-07536-t003:** FERAD scores from the database.

Variable	Number	Mean FERAD	*p* Value of FERAD
African American	244	19,509	NSS
Caucasian American	164	15,885	
Age < 60 years of age	206	20,519	NSS
Age > 60 years of age	206	15,084	
Sex male	376	19,540	0.052 *
Sex female	40	4262	
BMI < 28	187	16,830	NSS
BMI > 28	211	15,197	
No Diabetes	234	15,900	NSS
Diabetes	170	20,704	

A total of 52.1% males died versus 22.5% females OR 0.2666 (CI 0.1235–0.5754) *p* < 0.0004. Mean age in females was 52.48 ± 11.2 and in males, 62.42 ± 10.75 years; thus, age may explain mortality differences. NSS—not statistically significant. * Technically not significant, but the low number of females made the comparison unbalanced, so the power was decreased; however, the trend was strong.

**Table 4 ijms-24-07536-t004:** Differences in Survival.

Quartile Parameters	Quartile 1	Quartile 2	Quartile 3	Quartile 4
FERAD mean ± SD	6816 ± 7822	69,575 ± 132,844	41,880 ± 89,230	50,704 ± 114,298
Number total 445	111	111	112	111
% Males	80	99	96	92
Age mean ± SD years	60.3 ± 12.3	63.8 ± 11.1	62.2 ± 11.4	62.7 ± 11.4
Ethnicity %AA	58	62	62	62
BMI mean ± SD	27.9 ± 4.8	28.6 ± 5.9	27.7 ± 5.9	29.0 ± 6.0
Mean Effluent p87	0.070 ± 0.123	0.200 ± 0.333 *	0.163 ± 0.289	0.283 ± 0.441 **

* Student’s *t*-test quartile 1 versus 2, *p* < 0.007; ** *t*-test quartile 1 versus 4, *p* < 0.011. AA—African American; SD—standard deviation.

**Table 5 ijms-24-07536-t005:** Medication and Effects on the SARS-CoV-2 Virus.

Drug Class	Mechanism of Action	Receptor	COVID-19 Action
Angiotensin converting enzyme-1 inhibitors	Increase in ACE expression	Angiotensin converting enzyme 2 (ACE2)	Reduced susceptibility and lung protection [30]
Angiotensin II receptor blockers	Possible Increase in ACE expression	Angiotensin II receptor	Reduced susceptibility and lung protection [31]
Calcium Channel Blockers	Blocks Cav1.2 L-channel pores—external and internal surfaces	Cav1.2 L-channel pores	Antiviral effect [32,33]
Diuretics	Blockade of resorption of sodium/water block Cl-receptor channel	Tubule or connecting/loop/collecting duct vasopressin receptors	No effect [34]
Beta-blockers	Blocks macrophage catecholamine receptors	Alpha-1-adrenergic receptor	Beneficial effect in pneumonia by reducing hyperinflammation [35]

## Data Availability

Once database is finalized it can be made available for data-sharing.

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
