# Peer review of "In the SARS-CoV-2 Pandora Pandemic: Can the Stance of Premorbid Intestinal Innate Immune System as Measured by Fecal Adnab-9 Binding of p87:Blood Ferritin, Yielding the FERAD Ratio, Predict COVID-19 Susceptibility and Survival in a Prospective Population Database?"

_ijms, 2023, doi:10.3390/ijms24087536_

Round 1
Reviewer 1 Report
The significance of this problem is crucial for both people and globally. Assessment of the overall condition in people (with or without risk for severe COVID-19 disease) using given biomarkers and their use to predict a given condition is very important after this COVID-19 pandemic.
The overall strengths of the article are pretty obvious. Firstly, it helps readers even with entry-level knowledge for analysis of big data to understand the trends and opportunities in the research studies. Secondly, it illustrated general facts and provided readers with possible solutions. However, there are major points in the article which need refinement.
The introduction reveals what is already known about this topic, but the aim is not so clear.
The variables are well defined and measured appropriately. The study methods are valid and reliable.
The data is presented in an appropriate way. The results clearly presented and illustrated with tables and graphs. But the all of them are discussed from different angles without placed into specific context.
The conclusions do not fully answer the aim of the study and are not supported by own results.
The limitations of the study are not fatal, but must be written more clearly in one paragraph, because they are opportunities for future research.
Here are some suggestions that would improve the introduction:
1. According to the context in the Introduction, the authors can give some information about the innate immunity which is the first line of defense against SARS-CoV-2 and a scheme/figure of mechanism.
2. According to context, the authors can provide some information about other similar studies supported by relevant references.
3. Specify the graph titles (y-axis variable; x-axis) in Fig.2,Fig.6, Fig.10….
4. Fig.12 –Western blot of stool with specific p87 bands. - The protein marker should be labeled. How many repeats does the blot have? Has the results been statistically processed or is it just a representative blot? In the fresh sample (Wet), the intensity of the bands is much stronger than the samples from day 1 and day 3 (D1, D3). The sample from day 6 (D6) also has a stronger intensity? Is there an explanation in these results?
5. In several times in the manuscript, male and female individuals are mentioned. In chapter "Methods and materials", could you explain in more detail what necessitates the sex division?
6. There is a lack of references throughout the work, especially in the chapter "Discussion". Many studies are mentioned, but there are no references.
The article is written very detailed. The reader understands the importance of the topic although the large volume of scientific data.
All these refinements may make the article more interesting for readers from multiple backgrounds.
Author Response
Cover Letter Response to reviewers. Changes in the manuscript itself are color-coded, please see below.
Reviewer 1: Three categories were positive but 3 were in the “Can be approved” category and were:
- Are all the cited references relevant to the research?
- Are the results clearly presented?
- Are the conclusions supported by the results?
These are important issues which we will address in our response, “needing refinement” below.
Response to comments of Reviewer 1: (In Red)
General: We find the comments most insightful and the suggestions most useful and thank the reviewer for the in-depth appraisal which will hopefully lead to greater clarity and readability for the reader.
Specific: Each response will relate to the reviewer’s comment and suggestion and will inherently address the deficiencies outlines in the 3 deficient categories.
Reviewer 1 Comment 1: The introduction reveals what is already known about this topic but the aim is not so clear.
Reviewer 1 Suggestion 1: Authors can give some information about the innate immunity which is the first line of defense against SARS-CoV-2 and a scheme/figure of mechanism.
Response to Reviewer 1 Comment 1:
The aim has been sharpened by the shortening of the title as suggested by reviewer 3. We also now include an up to date citation not only of the innate immune system, but also of the expression of acute-phase reactants to provide a more complete picture of the panoply of response. This masterful review now cited as reference #15 which was not available at time of resubmission.
Reviewer’s 1 Comment 2: All tables and graphs are discussed from different angles without placed into specific context.
Suggestion 2: The authors can provide some information about other similar studies supported by the relevant references.
Response to Reviewer 1 Comment 2:
The data shown in the tables and graphs are entirely novel and as a result, other than a previously cited study in the same population (Table 2 and Figure 2), there are no comparable data from “similar” studies for comparison as yet, as the context has yet to be fully developed once the results have been reproduced by other researchers in the field.
Reviewer 1 Comment 3: The conclusions do not fully answer the aim of the study and are not supported by own results.
Suggestion 3: Specify graph titles (Y-axis variable; x axis) in Fig. 2, Fig.6, Fig, 10…
Response to Reviewer 1 Comment 3:
The aims of the study are not fully expressed and can be improved as suggested by the reviewer by refining the titles, axes in figures 2,3,and 10 which has been done. We did not mark in red the actual changes in the figures themselves as this may have resulted in difficulty in graphic reproduction.
Reviewer 1 Comment 4, regarding Fig. 12.
- The protein marker should be labeled.
- How many repeats does the blot have?
- Has the results been statistically processed or is it just a representative blot?
- The intensity of bands in the fresh sample (Wet) is much stronger than samples from days 1 and three. Day 6 also has a stronger intensity.
Response to Reviewer 1 Comment 4 regarding comment 4.
The Marker Lane has now been annotated as suggested. This was a single representative observational blot and not densitometry was performed. Understandably, the wet (fresh) sample would be the most intense as there was not degradation of the protein bands and thus was the strongest. The samples were left at room temperature subject to changes determined by air conditioning and humidity, beyond our control and this may have resulted in the subtle changes in the Day 6 sample detected by the astute reviewer.
Reviewer 1, Comment 5: Several times in the manuscript male and female individuals are mentioned.
Suggestion for e: In chapter “Methods and materials” explain in more detail what necessitates the sex division?
Response to Reviewer 1, Comment 5.
There are different responses and severity of COVID-19 disease in males and females and that is why we reported these wherever these data were available. This is now clearly enunciated in the relevant section. We have pointed out that only a small percentage of US Veterans are females and there is a great sensitivity to this fact in the Veterans Administration. Any data obtained on this demographic is therefore very important and hence our emphasis on this through the manuscript.
Reviewer 1 Comment 6:
There is a lack of references throughout the work, especially in the chapter “Discussion”. Many studies are mentioned but there are no references.
Response to Reviewer 1 Comment 6: We introduced a citation of Ferritins in renal insufficiency which was lacking. Another reference which was cited should have been placed earlier in the relevant section. Almost half the references cited appear in the discussion section which lends credence to our supposition that this section is fairly well-referenced.
Reviewer 2 Report
McVicker et al. contributed in manuscript entitled "In The SARS-CoV-2 Pandora Pandemic: Can the Stance of Premorbid Intestinal Innate Immune System as Measured by Fecal Adnab-9 Binding of p87:Blood Ferritin, yielding the FE-RAD Ratio, Allow Modeling to Predict COVID-19 Susceptibil-ity and Survival in a Prospective Population Database?" is well written and structured.
The work and results are original and have significant impact in COVID 19 research.
Author Response
Reviewer 2: Comment 1:
All categories were positive and the manuscript was characterized as: “well-written and structured”. It was also noted that: ”The work and results are original and have significant impact in COVID 19 research”.
Response to Comments of Reviewer 2.
We are grateful to the reviewer for the encouraging statements.
Reviewer 3 Report
Tobi et al in the article titled `In The SaRS-CoV-2 Pandora: Can the Stance of Premorbid Intestinal Innate Immune System as Measured by fecal Adnab-9 Binding of p87: Blood Ferritin, yielding the FE-RAD Ratio, Allow Modelling Prospective Population Database` describes a correlation COVID-19 prediction and severity of outcome in different populations based on FERAD ratio, using parameters such as p87 relevant for the colorectal neoplasia.
The article is well written and structured. The figures and tables are of good quality.
The authors studied a great amount of data available of various clinical databases. They describe in details how FERAD ration affects patients in terms of severity of COVID-19. They also discuss age, race and medication affect in this matter.
What could be improved would be the title. Though appear as clear, still it is long and a hard to follow.
Another matter is, maybe authors were supposed to compare these four groups they used with same groups of patients from different geographical locations and also to take in consideration other environmental factors that these populations are exposed to.
Author Response
Cover Letter Response to reviewers. Changes in the manuscript itself are color-coded, please see below.
Reviewer 3: All categories were positive but some interesting comments for improvement were:
Comment 1: What could be improved would be the title.
Response to Comments of Reviewer 3. (in gray)
We appreciate the thoughtful remarks and agree with the first. To better focus the title as suggested we have deleted the words “Allow Modeling”.
Comment 2: Maybe authors were supposed to compare these four groups with same groups they used from different geographical locations and also to take in consideration other environmental factors.
Response to Comment 2: This comment is very interesting. We would have very much liked to do this by including different patients from different locations. Given that our database was limited to patients from one location we could not access other locations. However, in order to somewhat compensate for this, we presented data of the effect of medications on COVID-9. We drew from 5 international study results from the literature summarized in Table 5 to address this important topic. Each citation originated from a distinct geographic area. In future studies we will make every effort to include such data for comparison.
Round 2
Reviewer 1 Report
Dear Authors,
I have been carefully reviewed your revised article with the id number " ijms-2284105". In my opinion, this revised article incorporates all of the points raised in the original draft to the best of my knowledge.
This article will contribute significantly the prognostic data and the ability to predict disease severity using different potential prognostic biomarkers. The InImS effect on both disease susceptibility and outcome, in both the child and adult, including the effects of underlying disease is intriguing.
Best wishes to all of the authors who contributed to the production of this wonderful work and congratulations on their future endeavors.